# Recent Updates and Advances in the Use of Glycated Albumin for the Diagnosis and Monitoring of Diabetes and Renal, Cerebro- and Cardio-Metabolic Diseases

**DOI:** 10.3390/jcm9113634

**Published:** 2020-11-11

**Authors:** Rosaria Vincenza Giglio, Bruna Lo Sasso, Luisa Agnello, Giulia Bivona, Rosanna Maniscalco, Daniela Ligi, Ferdinando Mannello, Marcello Ciaccio

**Affiliations:** 1Department of Biomedicine, Neuroscience and Advanced Diagnostics, Institute of Clinical Biochemistry, Clinical Molecular Medicine and Laboratory Medicine, University of Palermo, 90121 Palermo, Italy; rosariavincenza.giglio@unipa.it (R.V.G.); bruna.losasso@unipa.it (B.L.S.); luisa.agnello@unipa.it (L.A.); giulia.bivona@unipa.it (G.B.); 2Department of Laboratory Medicine, University Hospital Paolo Giaccone, 90127 Palermo, Italy; 3Department of Biomolecular Sciences, Section of Biochemistry and Biotechnology, University Carlo Bo of Urbino, 61029 Urbino, Italy; r.maniscalco@campus.uniurb.it (R.M.); daniela.ligi@uniurb.it (D.L.)

**Keywords:** glycated albumin, diabetes, dyslipidemia, obesity, kidney disease, therapy, cardiovascular disease, cerebrovascular disease

## Abstract

Diabetes mellitus is a heterogeneous and dysmetabolic chronic disease in which the laboratory plays a fundamental role, from diagnosis to monitoring therapy and studying complications. Early diagnosis and good glycemic control should start as early as possible to delay and prevent metabolic and cardio-vascular complications secondary to this disease. Glycated hemoglobin is currently used as the reference parameter. The accuracy of the glycated hemoglobin dosage may be compromised in subjects suffering from chronic renal failure and terminal nephropathy, affected by the reduction in the survival of erythrocytes, with consequent decrease in the time available for glucose to attach to the hemoglobin. In the presence of these renal comorbidities as well as hemoglobinopathies and pregnancy, glycated hemoglobin is not reliable. In such conditions, dosage of glycated albumin can help. Glycated albumin is not only useful for short-term diagnosis and monitoring but predicts the risk of diabetes, even in the presence of euglycemia. This protein is modified in subjects who do not yet have a glycemic alteration but, as a predictive factor, heralds the risk of diabetic disease. This review summarizes the importance of glycated albumin as a biomarker for predicting and stratifying the cardiovascular risk linked to multiorgan metabolic alterations.

## 1. Introduction

In the context of precision medicine, it would be desirable to estimate the “cardiometabolic risk” of each individual by evaluating the presence of risk factors which could influence the metabolism leading to increased cardiovascular mortality [1,2,3].

Glycated Albumin (GA) [4,5,6] represent a useful tool reflecting glycemic homeostasis over a period of three weeks. GA provides a qualitatively and quantitatively significant contribution to the clinical process of screening, prediction, prevention, diagnosis, prognosis and monitoring of the metabolic disorders of several organs [7,8,9].

GA, a unique post-translational product of the glycosylation of albumin, has been shown to be a useful biomarker in: (i) clinical conditions in which the dosage of glycated hemoglobin (HbA1c) does not reflect glycemic compensation, such as chronic renal failure, gestational diabetes, dialysis complications, hemorrhages, recent transfusions, hemolytic anemia and hemoglobin variants [6,9,10,11,12,13]; (ii) patients with poorly controlled diabetes and with postprandial hyperglycemia in order to monitor short-term glycemic fluctuation [14,15]; (iii) diabetic patients for monitoring therapeutic response [16,17,18,19]; (iv) detecting prediabetes even in the presence of normoglycemia, which is a priority because the effectiveness of this lifestyle intervention is lost if diabetes has already developed [20,21]. It is well known that fructosamine represents all the circulating glycated proteins, therefore Glycated Albumin is biochemically recognized as a form of fructosamine [14]. With the Oral Glucose Tolerance Test (OGTT) as the diagnostic standard [11,22], GA has been shown to be superior to fructosamine alone in detecting hyperglycemic states [8,23]. Moreover, the GA/HbA1c ratio is very important from a diagnostic point of view to detect patients with postprandial hyperglycemia or large glycemic excursion [15,24].

The aim of this review is to describe GA’s contribution in diabetes and metabolic conditions and to present its potential usefulness for predicting and stratifying cardiometabolic risk.

## 2. Search Strategy

Using electronic databases [MEDLINE (1975–2020), EMBASE and SCOPUS (2000–2020)] and abstracts from national and international meetings, we searched for interventional and observational studies in humans which assessed the association between glycated albumin and cardiometabolic risk factors. A total of 74 studies were included in the present review, extracted using combinations of the following keywords: glycated albumin, diabetes, dyslipidemia, obesity, kidney disease, cardiovascular risk, and therapy.

## 3. Glycated Albumin (GA)

Albumin is the most abundant plasma protein, with a half-life of about 21 days, that undergoes non-enzymatic glycosylation due to its high circulating concentration, the number of lysine and arginine residues making it more prone to glycation [6,25].

The protein consists of 585 amino acid residues organized in a single polypeptide chain stabilized by 17 disulfide bridges and comprising three homologous domains (I, II, and III) assembled to form a heart-shaped molecule. Each domain is further organized into two subdomains (A and B), which share analogous structural motifs. The glycation-induced modifications significantly alter the structure of the two subdomains of the second domain. These two subdomains act as binding sites, with preferential activity for different endogenous substrates (such as bilirubin and porphyrins), and exogenous compounds (such as benzodiazepines and ibuprofen).

The glycation process also modifies the *n*-terminal region and the Cysteine (Cys)-34 of albumin. Both regions are involved in the binding of drugs and various metal cations. GA can contain both early and advanced glycation products [7,26]. The latter can derive both from the transformation of early adducts already present at the protein level, and from new interactions between the protein and preformed early glycation products [27]. Both the reduction of circulating levels and structural changes capable of modifying biochemical and physical properties of the protein can alter the binding properties of albumin against multiple drugs, and reduce its antioxidant and scavenger power [28,29] (Figure 1). Therefore, the glycation process of albumin causes potential changes in the therapeutic and/or toxic effects of all the drugs that bind albumin, with a significant variation mainly for agents acting in a relatively narrow therapeutic window [30]. It is possible that the reduced binding capacity towards fatty acids [31], especially arachidonic acid, is associated with an increased thrombotic risk in patients with metabolic syndrome and diabetes [32,33].

### GA Test

The American Diabetes Association (ADA) and the European Association Diabetes Study (EASD) recommend “patient-centered” management of glycemic control in patients with Type 2 Diabetes Mellitus (T2DM) and the selection only of biomarkers, such as GA, that reflect the individual health status of the diabetic patient, maintaining the balance between risks and benefits [34].

The GA test provides information on the mean blood glucose concentration in the last 15–20 days before blood collection [4,6,8,11,19,24].

The GA can be measured by several methods, including affinity chromatography, ion exchange, high-performance liquid chromatography and immunoassays (such as enzyme-linked immune-sorbent assays and radio-immunoassays) (as reviewed in [35,36]). GA can be measured on both serum and plasma, where it is stable both for 24 h at room temperature, or at 4 °C for a week, or at −80 °C for several months [5,34].

Recently, an innovative and very promising electrochemical immunoassay using nanozymes has also been developed, showing good linearity and a lower limit of detection [37].

Interestingly, an enzymatic method that shows good analytical performance has also recently been automated (Lucic^®^ GA-L kit, Asahi Kasei Pharma Corporation, Tokyo, Japan). The method is based on the initial elimination of endogenous glycated amino acids and peroxides mediated by a ketamine oxidase and a peroxidase. The GA is then hydrolyzed by an albumin-specific proteinase, and the products of this reaction are oxidized by a ketamine oxidase. The hydrogen peroxide produced is then measured quantitatively by the classic colorimetric method of Trinder. Meanwhile, in parallel, the concentration of albumin is measured by the bromocresol violet method, allowing the results to be expressed as the ratio between GA and total albumin [38,39]. The result of GA is provided as a percentage (GA%) of total albumin. The GA Upper Reference Limit (URL) of 14.5% (95% CI: 14.3–14.7) has been established in Caucasian healthy subjects [5].

## 4. GA and Cardio-Metabolic Risk Factors

High GA levels can induce irreversible damage in several organs, which represent the main targets for complications of diabetes mellitus (e.g., the coronary arteries, the cardiovascular system, the kidneys, the thyroid, the skeletal muscles, the eyes, and the central nervous system).

For example, in the kidney, GA stimulates epithelial and mesangial cells to produce pro-oxidant molecules and connective tissue, thus contributing to the onset of nephropathy [40,41,42,43].

In the field of cardiovascular diseases, GA exerts pro-inflammatory and pro-oxidant effects on cardiomyocytes [44], playing a role in the activation and aggregation of platelets [32], and also regulating the expression of adhesion molecules involved in the formation of atherosclerotic plaques, such as the InterCellular Adhesion Molecule (ICAM)-1 and the Vascular Cell Adhesion Molecule (VCAM)-1 [45], but also promoting deleterious oxidation [46].

Cells of the skeletal muscle are also sensible to the increased concentrations of GA, which stimulate an over-expression of inflammatory cytokines [47].

Furthermore, a significant association between thyroid hormones and GA in both euthyroid and subclinical hypothyroid individuals has recently been described [48].

At the metabolic level, the increased intracellular production of reactive oxygen species (ROS) may reduce the secretion of insulin [49]. GA exerts its harmful effects by the activation of the cellular receptor for advanced glycation end products (RAGEs). Indeed, the activation of RAGEs stimulates the production of pro-inflammatory cytokines, and induces apoptosis, oxidative stress, and platelet aggregation, events that have been associated with the increase of AGE levels and GA levels [50].

In the following subparagraphs, the correlation between GA and cardiometabolic risk factors such as diabetes, dyslipidemia, obesity, kidney disease and cerebro-cardiovascular diseases will be described in detail (Figure 2).

### 4.1. GA and Diabetes

The pathogenesis of T2DM is linked to insulin-resistance, which leads to a slow decline of pancreatic β-cells; both pathological states influence each other and, presumably, synergistically exacerbate diabetes inducing the failure of insulin over time [51].

In addition to the use of HbA1c for the diagnosis of diabetes [52], the use of additional glycated proteins for this diagnosis and its complications has been widely recommended [11].

The glycation rate of albumin is greater than that of hemoglobin and would reflect early hyperglycemia before HbA1c [8,53].

#### 4.1.1. GA and Diabetes Diagnosis

As a medium-term glycemic status indicator, GA is originally taken into consideration for the diagnosis of diabetes in patients with hemoglobinopathies and other conditions in which the measurement of HbA1c is not reliable [4,6,52,54]. It has also been assessed in screening programs to detect individuals with prediabetes [23,25]. It is necessary to discern the various conditions related to diabetes and to assess the value of GA based on the presence of comorbidity [35]. For example, Koga et al. reported that serum GA levels are significantly lower than HbA1c in diabetic patients with preserved insulin secretion [55,56,57].

The homeostasis model assessment of β-cell function (HOMA-β), an index of insulin secretory function, is negatively correlated with GA, but is not correlated with HbA1c. Indeed, it has been assumed that this phenomenon is due to the association between reduced insulin secretion and increased glucose excursions, including postprandial hyperglycemia [49].

#### 4.1.2. GA and Monitoring of Diabetes Treatment

The evaluation of the efficacy of therapy in diabetic patients is mandatory in order to avoid glycemic failure and its consequences.

GA levels may be useful in predicting the patient’s response to hypoglycemic therapy. Lee et al. performed a study on Korean patients with poorly controlled insulin-naive T2DM. These subjects were randomized based on obesity and the dose of glimepiride with a 1:1 ratio of insulin detemir (prolonged-action analog) and Biphasic Insulin (rapid-action analog) [58]. Mean HbA1c, GA, fasting and stimulated plasma glucose levels were significantly reduced after 16 weeks [58]. GA decreased more rapidly than HbA1c during intensive insulin therapy. Thus, GA has been considered a useful biomarker for monitoring short-term variations of glycemic control during the treatment of diabetic patients [58,59].

GA could also be potentially useful for monitoring the beneficial effects of nutraceuticals. Li et al. performed a double-blind, randomized, placebo-controlled study to evaluate fiber supplementation during a 12 week integration period [60]. All glucose metabolism biomarkers, including HOMA-Insulin Resistance, HbA1C and GA, improved after 12 weeks.

### 4.2. GA and Dyslipidemia

#### 4.2.1. GA and Dyslipidemia Diagnosis

Dyslipidemia is a clinical condition frequently found in diabetic patients. It is characterized by the alteration of lipid metabolism. Specifically, diabetes is associated with hypertriglyceridemia due to increased levels of very low density lipoprotein (VLDL), which are enriched by Apo-CIII (apolipoprotein c-III). Indeed, patients with T1DM and T2DM have increased ApoC-III levels [61]. It has been shown that the inhibition of ApoC-III can improve dyslipidemia in diabetic patients [62]. The underlying mechanism could be the improvement of lipoprotein lipase (LPL) activity and the increased hepatic uptake of triglyceride-rich lipoprotein (TRL) [62]. A strong relationship between improved insulin sensitivity and ApoC-III suppression has been described [62,63]. Park et al. found that patients with GA > 17.0% showed a direct and significant correlation with fasting plasma glucose (FPG), TC (total cholesterol), TG (triglyceride), and LDL-C (low density lipoprotein-cholesterol), as compared to patients with GA ≤ 17.0% [63]; these findings indicate that GA can be used as a tool for screening high-risk diabetic patients for early diagnosis of dyslipidemia.

#### 4.2.2. GA and Monitoring of Dyslipidemia Treatment

Patients with dyslipidemia who do not reach the LDL-C target via lifestyle changes only or with nutraceuticals undergo hypocholesterolemia therapy, such as statins. Despite the numerous beneficial effects of statins, especially atorvastatin and rosuvastatin, on LDL levels [64,65,66,67], some collateral effects, collectively known as statin intolerance (SI) syndrome, have been reported. Specifically, SI syndrome is characterized by the increase of creatine-phosphokinase (CPK) and other bio humoral biomarkers (such as lipase and amylase) as well as glucose homeostasis alterations, which can lead to the development of diabetes [68,69,70,71,72]. GA could represent a useful biomarker for monitoring the effect of hypocholesterolemia treatment on glucose homeostasis, as shown by some authors [73]. A randomized, double-blind, placebo-controlled study was conducted in adult patients with T2DM and hypertriglyceridemia treated with metformin and volanesorsen, respectively [74]. Volanesorsen improved insulin sensitivity, which was associated with a decrease in GA and HbA1c levels [74]. These data suggest that GA could be used for monitoring the efficacy of lipid-lowering therapies [63].

### 4.3. GA and Obesity

The World Health Organization (WHO) defines obesity as ‘‘abnormal or excessive fat accumulation that represents a risk to health’’. A BMI (body mass index) > 29.9 kg/m^2^ identifies obese patients. More in-depth research has consistently revealed that HbA1c is positively associated with BMI, while GA is negatively associated with BMI [75]. Recently a study exploring the association of waist circumference (WC) with GA and to assess the extent to which WC influences GA showed that GA levels and GA/HbA1c ratio were further decreased in subjects with central obesity than those without, suggesting that WC was a significant negative determinant of GA [76]. Thus, GA could not be a reliable biomarker of glucose homeostasis in obese subjects. Similarly, Sumner et al. found that, in obese subjects, the HbA1c was a much better diagnostic test than the GA [77] and the combination of the tests did not improve the diagnostic sensitivity [77,78].

In a study including subjects with normal and reduced glucose tolerance, BMI was found to be negatively associated with GA and GA/HbA1c ratio. Interestingly, the inflammatory status was associated positively with BMI and negatively with GA [79], confirming data previously described [80,81].

Several studies assessed the negative relationship between BMI and GA in the diabetic population [81,82,83], revealing a negative relationship between BMI and GA in diabetes.

Although a correlation between BMI and the GA/HbA1c ratio has been reported [80], contradictory studies have showed that the reduction of BMI is associated with a high GA/HbA1c ratio [15], whereas other studies reported that there is an inverse association between BMI and GA/HbA1c ratio in subjects with various states of glucose tolerance [84]. Thus, further studies are mandatory in order to clarify the relationship between BMI and GA/HbA1c ratio.

The relationship between BMI and GA levels in diabetic and non-diabetic patients has been evaluated [83], revealing that a decreasing trend in GA levels and GA/HbA1c ratio is associated with an increasing trend in BMI, suggesting that BMI has a negative effect on the GA/HbA1c ratio [83]. In a wide study, the association between serum GA and anthropometric variables revealed that patients with BMI > 25.0 kg/m^2^ had significantly higher HbA1C and body fat parameters, but significantly lower GA levels (*p* < 0.05) [75].

Overall, the evidence supports a negative relationship between BMI and GA, regardless of the presence of diabetes [80,81,83]. The negative relationship between GA and BMI could be due to the inverse correlation between BMI and total albumin. Indeed, obese subjects present chronic inflammation, characterized by an increased release of cytokines from adipose tissue (adipocytokines), which promote the synthesis of CRP in the liver [85] by reducing the synthesis rate and increasing the catabolic rate of the albumin [86]. That GA is a ketamine formed through the non-enzymatic glycation reaction of serum albumin partly explains the relatively low concentration of GA in obese patients [79,87,88].

### 4.4. GA and Kidney Diseases

Chronic kidney disease (CKD) is associated with insulin resistance and, in advanced CKD, decreased insulin degradation. Thus, monitoring glycemic homeostasis in patients with CKD is fundamental [89]. HbA1c remains the gold standard with CKD patients. However, some authors have evaluated the role of GA in CKD patients [90].

#### 4.4.1. Glycated Albumin and Kidney Disease Diagnosis

GA is associated with albumin turnover and it is still unclear whether proteinuria affects GA values in diabetic patients with CKD [9]. Patients with advanced diabetic nephropathy and diabetic patients with non-diabetic kidney disease exhibit evident and significant proteinuria. GA would appear to be more accurate in evaluating glycemic control than HbA1c in patients with advanced CKD [91].

Patients affected by nephrotic syndrome have an increase in albumin synthesis and in the fractional catabolic rate, with consequent rapid albumin turnover [92,93]. Kidney function may not directly affect GA values; however, higher levels of GA are associated with all-cause or cardiovascular mortality in diabetic patients on hemodialysis [94].

The interpretation of GA may be limited in the context of CKD due to proteinuria and impaired protein homeostasis and half-life [95]. The associations of HbA1c and GA with FPG showed a weak correlation in dialysis patients compared to those not on dialysis [96], suggesting a recommendation for GA, compared to HbA1c, for monitoring glycemic control in dialysis patients [8,97].

A study of diabetic patients with CKD revealed a significant correlation between GA and UPE [98].

In the Atherosclerosis Risk In Communities (ARIC) study, the association of HbA1c and GA with FPG in different CKD categories revealed significant correlations of HbA1c and GA with FPG, with lower levels in patients with severe chronic renal failure and very severe CKD [99].

A strong association of GA with retinopathy and with the risk of CKD and incident diabetes was found, highlighting that the prediction of CKD by GA was almost as strong as by HbA1c [99,100].

Interestingly, the association between GA and arterial stiffness in CKD patients was analyzed [101], revealing that patients not suffering from chronic renal failure but with arterial stiffness showed higher GA levels than those without arterial stiffness, and that GA correlated significantly with both arterial stiffness and FPG, while HOMA-IR (homeostatic model assessment-insulin resistance) did not show any significant correlation with arterial stiffness.

The association between GA and the progression of Diabetic Nephropathy (DN) revealed that mean GA levels were more closely associated with DN progression than HbA1c in T2DM patients [102].

In conclusion, serum GA is a potentially useful glycemic index in diabetic patients with chronic renal failure, as it is not affected by anemia and associated treatments [102,103,104].

#### 4.4.2. Glycated Albumin and Kidney Disease Treatment Monitoring

Drug therapy is different in the various stages of the disease. Specifically, in the initial stages it is usually possible to administer therapy for the contrasting causes underlying the nephropathy. In the advanced stages, it is necessary to adopt treatments to control complications of renal failure by “poly-therapy” [89].

The Japanese Society for Dialysis Therapy recommended GA levels < 20.0% as a target for glycemic control in patients without a history of cardiovascular events and GA level < 24.0% as a provisional target for patients with a history of cardiovascular events or with a tendency to hypoglycemic episodes [105].

### 4.5. GA and Cerebro-Cardiovascular Diseases

There are various causes that determine cerebrovascular diseases, including atherosclerosis, a pathological condition characterized by arterial inflammation and accumulation of cholesterol, leading to the formation of thick plaques [106].

There is a strong correlation between GA and atheromatous plaque, as demonstrated by numerous studies [101,107,108,109,110,111,112,113,114,115,116,117].

#### 4.5.1. GA and Cerebro-Cardiovascular Diagnosis

A relationship among molecular (GA, HbA1c and CRP) and vascular (Carotid Intima Media Thickness (CIMT)) biomarkers has been reported. Increased levels of GA have been associated with increased levels of CRP and carotid atherosclerosis, especially when considering diabetic patients with proven cardiovascular diseases [107]. Since GA levels have been proven to be independent predictors of the presence of Coronary Artery Disease (CAD), GA can generally be used for decision making after initiation or modification of therapy [109].

GA was found to be a more useful predictor for the presence of microangiopathy than HbA1c [100,110], probably because GA is more closely related to glycemic fluctuation and excursion than HbA1c in diabetic patients with poor glycemic control. Noteworthy, postprandial hyperglycemia contributes more to cardiovascular events and the risk of death than fasting hyperglycemia [118,119]. GA represents a more reliable indicator of the glycemic status than HbA1c in older subjects [120].

Recently, GA has been found to play a role in the increased arterial stiffness observed in diabetic patients, due to the activation of Nuclear Factor kappa-light-chain-enhancer of activated B cells (NF-kB) in endothelial cells and the proliferation of vascular smooth muscle cells [121], regulating also Nicotinamide Adenine Dinucleotide Phosphate Oxidase (NADPH) oxidase and inducing sustained production of ROS (reactive oxygen species) in human endothelial cells [122]. In this context, GA stimulates Transforming Growth Factor beta (TGF-b) and increases oxidative stress, inducing vasculopathy in experimental models [123]. 

GA has been associated with CIMT in patients with T2DM without any CAD and Peripheral Artery Disease (PAD) [119], as well as in the general population [115].

Although further studies are required to clarify the mechanisms involved in the relationship between GA and atherosclerosis, a possible cut-off value of GA for the prediction of CAD in patients with diabetes was defined at 17.9% (sensitivity 0.639 and specificity 0.639) [119].

Recently, it has been highlighted that pre-stroke glucose variability estimated by GA was significantly associated with an increased risk of severe initial stroke severity and large infarct volume in acute ischemic stroke patients with diabetes mellitus [124], suggesting that GA could be a useful tool for discerning/identifying early cerebrovascular disease.

In patients undergoing coronary artery Computed Tomography Angiography (CCTA), [116] the CAD group showed significantly higher GA and HbA1c levels than the non-CAD group (*p* < 0.05) with a positive correlation between GA and HbA1c (*r* = 0.551, *p* < 0.0001); in particular, GA (*r*: 1.30, *p* = 0.02) was the only predictor of the presence of CAD in the diabetes mellitus group [116].

#### 4.5.2. GA and Cerebro-Cardiovascular Treatment Monitoring

The Clopidogrel in High-risk Patients with Acute Non-disabling Cerebrovascular Events (CHANCE) trial analysed 3044 patients with minor ischemic stroke or Transient Ischemic Attack (TIA) from 73 predetermined clinical sites, who underwent measurement of baseline GA levels. Patients were divided into two groups based on a GA level of 15.5%, considered the threshold value for the development of diabetes [117]. The primary outcome was a stroke recurrence over 90 days of follow-up. Patients with minor ischemic stroke or TIA were randomized to antiplatelet therapy with Clopidogrel plus AcetylSalicylic Acid (ASA) or only ASA. A significant interaction of GA levels was found with the two antiplatelet therapy groups after adjustment for age, gender and other conventional confounding factors (*p* = 0.009) and the interaction remained constant after further correction for diabetes history (*p* = 0.010). In patients with the lowest GA levels, a stroke occurred in 5.5% of patients in the clopidogrel + ASA group and in 12.7% in the ASA only group (HR corrected: 0.40; 95% CI: 0.26–0.61; *p* < 0.001) [117]. In addition, in patients with elevated GA levels, stroke occurred in 9.2% of patients in the clopidogrel + ASA group and 11.4% in the ASA only group (HR corrected: 0.79; 95% CI: 0.60–1.05; *p* = 0.103). Basal GA levels may predict the effect of ASA alone or of double antiplatelet therapy on the risk of stroke recurrence, ischemic stroke and combined vascular events in acute patients with minor stroke or high-risk TIA [117].

## 5. Discussion and Conclusions

The present review showed that, in some specific clinical conditions, GA offers significant advantages for the monitoring of glycemic status compared to the currently available biomarkers used in the clinical setting.

However, in some specific pathological conditions, especially those characterized by changes in albumin metabolism (e.g., with increased albumin metabolism, such as in nephrotic syndrome and in hyperthyroidism), the value of GA may underestimate the real value of the glycemia. Specifically, in patients with kidney disease, the value of GA is reliable if the concentration of albumin in the urine does not exceed 3.5 g/24 h [98,125].

Other alterations of GA due to chronic inflammation could be observed in smokers, in patients with non-alcoholic liver disease, hypertriglyceridemia and hyperuricemia [126,127,128]. On the other hand, in all conditions in which albumin metabolism is reduced (such as cirrhosis and hypothyroidism), the GA levels are higher than the real mean glycemia and show some limitations in the monitoring of the glycemic status due to the reduced synthesis of albumin [129]. Above all, in subjects with cirrhosis, the GA/HbA1c ratio correlates with the liver function, allowing an indirect evaluation [130].

Although literature evidence supports the use of GA as a reliable biomarker for the assessment of glycemic status, there is no consensus on the decisional values to use for different clinical conditions (Table 1).

In conclusion, GA has all the potential to be an excellent predictor of cardio-metabolic risk, if evaluated in its fluctuations, compared to the metabolic parameters. However, additional studies are needed to introduce GA in clinical practice.

## Figures and Tables

**Figure 1 jcm-09-03634-f001:**
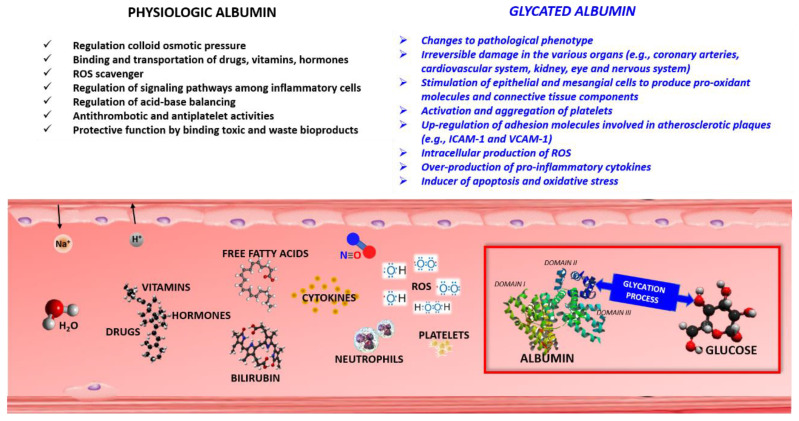
Schematic representation of the main physiological functions of albumin and pathological actions after specific glycation processes; ROS: reactive oxygen species; N = O: nitric oxide; ICAM-1: intercellular adhesion molecule-1; VCAM-1: vascular cell adhesion molecule-1.

**Figure 2 jcm-09-03634-f002:**
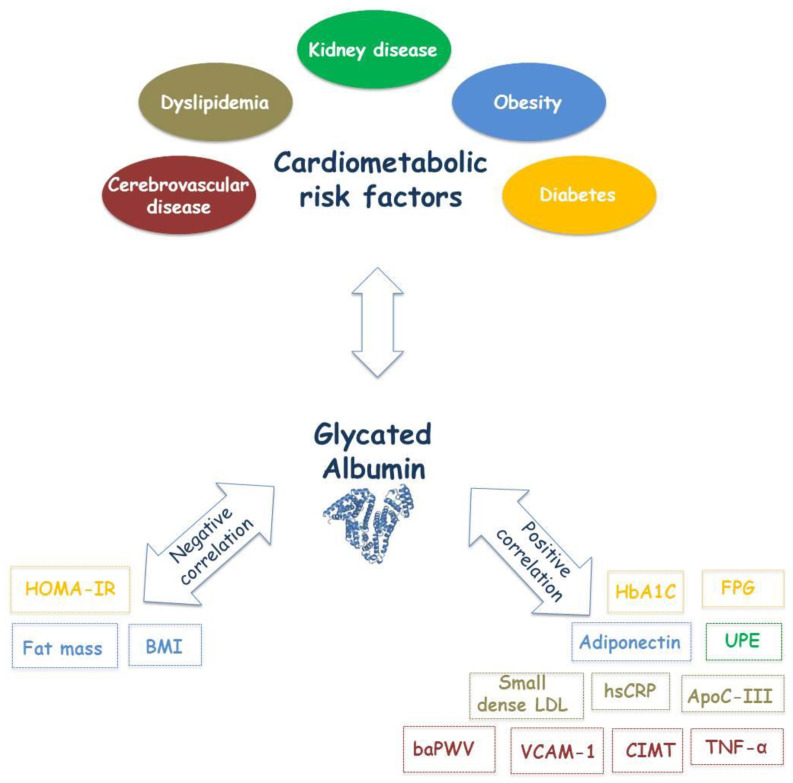
Schematic diagram of GA correlations with metabolic and vascular parameters. The cited parameters are divided into two groups based on their negative or positive correlation with GA and are reported in the same color as the risk factor to which they refer. (baPWV: brachial-ankle Pulse Wave Velocity; BMI: Body Mass Index; CIMT: Carotid Intima-Media Thickness; FPG: Fasting Plasma Glucose; GA: Glycated Albumin; HbA1C: Hemoglobin A1c; hsCRP: high sensitivity C Reactive Protein; LDL-C: Low-Density Lipoprotein Cholesterol; TNF-α: Tumor Necrosis Factors-α; UPE: Urinary Protein Excretion; VCAM-1: Vascular Cell Adhesion Molecule-1; HOMA-IR: homeostatic model assessment-insulin resistance; ApoC-III: apolipoprotein C-III).

**Table 1 jcm-09-03634-t001:** Human clinical studies on GA cut-off values in the diagnosis and monitoring of therapies of the main cardiometabolic risk factors.

Metabolic Risk Factors	Patients Number and Characteristics	Diagnosis Cut-Off Value of GA	Monitoring Therapy Cut-Off Value of GA	GA Parameters Associated (*r* = Correlation Coeff.)	Ref.
Diabetes Mellitus	1294 prediabetic patients	>14.9% for diagnosing Diabetes			[57]
Diabetes Mellitus	120 diabetic treated with SU		GA > 20% to switch to insulin		[58]
Dyslipidemia	102 newly diagnosed T2DM 263 diabetic and dyslipidemic patients	>15.6% for diabetic screening		HbA1C LDL-C FPG, TG and LDL-C ↓ TC (*r* = 0.012) LDL-C (*r* = 0.073)	[63]
Obesity	236 healthy non obese and obese individuals	>13.77% for detecting prediabetes		BMI (*r* = 0.24)	[77]
Kidney Disease	90 diabetic hemodialysis patients	>25% for predicting mortality			[94]
Kidney Disease	Without a history of cardiovascular events	<20% for glycemic control			[105]
	With a history of CV events or tendency to hypo-glycemic episodes	<24% for glycemic control			
Cerebro-cardiovascular disease	30,000 diabetic and obese subjects	>17.9% for the prediction of CAD			[114]
Cerebro-cardiovascular disease	1575 individuals from general population	>15.5% for predicting diabetes		HbA1c, hsCRP and CIMT	[115]

BMI: Body Mass Index; CAD: Coronary Artery Disease; CIMT: Carotid Intima Media Thickness; FPG: Fasting Plasma Glucose; CV: Cardiovascular; GA: Glycated Albumin; HbA1C: Hemoglobin A1c; hsCRP: high sensitivity C Reactive Protein; LDL-C: Low Density Lipoprotein-Cholesterol; SU: Sulphonyl-urea; T2DM: Type 2 Diabetes Mellitus; TC: Total Cholesterol; TG: Tri-Glycerides.

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
