# Peer review of "Recent Updates and Advances in the Use of Glycated Albumin for the Diagnosis and Monitoring of Diabetes and Renal, Cerebro- and Cardio-Metabolic Diseases"

_jcm, 2020, doi:10.3390/jcm9113634_

Round 1

Reviewer 1 Report

Glycated Albumin and Metabolic Risk Factors on 2 Renal, Cerebro- and Cardio-Vascular Diseases: 3 Updates and Advances.

Comments to the Authors:

The authors have done a good job in addressing the previous comments. Overall the manuscript reads well and is well organized. There are couple of minor concerns still that need to be addressed which are mentioned below.

  1. Section 4.3 (GA and Obesity), section 4.4.1 (Glycated albumin and kidney disease diagnosis) and section 4.5.1 (GA and cerebro-cardiovascular diagnosis) have lot of details about the studies that support the authors statements in respective sections. The authors need to minimize the details and just summarize the results that will help the readers.

Apart from the above mentioned comments the overall manuscript seems good.

Author Response

On behalf of all Authors, I would like to thank the expert Reviewer for the gratifying words: this is much appreciated.

For what concerns the minor concerns, we have minimized the details in sections 4.3, 4.4.1 and 4.5.1, as requested summarizing the results and improving finally the readibility.

Thanks again for the efforts, the attention and the time spent on our work.

Prof. F Mannello

Reviewer 2 Report

The review by Giglio et al. is a comprehensive review of the value of glycated albumin in the diagnosis and monitoring of diabetes and cardiometabolic disease.

However there are several recommendations:

  • The title should be condensed and contain the word the “diabetes” which is the main use of glycated albumin. A title which could be considered would be: Recent Updates and Advances in the use of Glycated Albumin for the Diagnosis and Monitoring of Diabetes and Cardiometabolic Disease
  • The text would benefit from some refinement in the language (perhaps an editor should go through and streamline the sentences).
  • Glycated albumin assays and calculations are well described but in addition, the fact that glycated albumin is a form of fructosamine should be added. Fructosamine represents all circulating glycated protein. And therefore glycated albumin is a form of fructosamine.
  • The figures are very good but the red print in Figure 1 should be changed to blue. The red print is very hard to read and will be impossible for anyone with red-green color blindness to read.

Author Response

First of all, I would like to thank the expert Reviewer for the insightful comments and suggestions, focused mainly to improve readibility and sound of our ms: this is much appreciated.

For what concenrs the recommendations, on behalf of all Authors, I changed the title according to the right suggestion, improving throughout the ms also phrases and sentences.

I added the details about the GA as fructosamine as suggested, and finally I have modified both the figure 1 and the figure of graphical abstract, improving the readibility changing the colors.

So, I appreciated very much your efforts and attentions focused to improve our ms.

all the best

Prof. F Mannello

This manuscript is a resubmission of an earlier submission. The following is a list of the peer review reports and author responses from that submission.

Round 1

Reviewer 1 Report

Major comments:

This manuscript tries to summarize the updates and contribution of glycated albumin as a biomarker for predicting and stratifying the cardiovascular risk linked to multiorgan metabolic conditions. The authors comprehensively reviewed large amounts of previous work to describe the advantages of glycated in clinical diagnosis and treatment for diabetes, dyslipidemia, obesity, kidney disease and cerebro-cardiocascular disease. However, I found lots of citation did not match the description of the text. For example, line 181-184, the study performed by Lee et al was described in reference 59 not 58; line 194-196, the study performed by Li et al was described in reference 61 not 60; line 337-345, the study performed by Xu et al was described in reference 91 not 90; Line 346-350, the study performed by Zemlin et al was describe in reference 58 not 57; line 351-362, the study performed by Sumner et al was described in reference 92, 93 not 91…and many inconsistencies can’t be listed them all. I suggest the authors proofread their manuscript more carefully and correct all the mistake thoroughly in the text.

Minor comments:

In figure 2, the authors used multiple circles to represent different cardiometabolic risk factors that related with GA. Does the size of the circle indicate the importance of GA in correlating with different risk factors? If yes, please provide detailed description with supporting reference. If not, please change the circle to the same size.

Therefore, I suggest accept this manuscript after major revision.

Author Response

REVIEWER #1:

COMMENT 1:

This manuscript tries to summarize the updates and contribution of glycated albumin as a biomarker for predicting and stratifying the cardiovascular risk linked to multiorgan metabolic conditions. The authors comprehensively reviewed large amounts of previous work to describe the advantages of glycated in clinical diagnosis and treatment for diabetes, dyslipidemia, obesity, kidney disease and cerebro-cardiovascular disease. However, I found lots of citation did not match the description of the text. For example, line 181-184, the study performed by Lee et al was described in reference 59 not 58; line 194-196, the study performed by Li et al was described in reference 61 not 60; line 337-345, the study performed by Xu et al was described in reference 91 not 90; Line 346-350, the study performed by Zemlin et al was describe in reference 58 not 57; line 351-362, the study performed by Sumner et al was described in reference 92, 93 not 91…and many inconsistencies can’t be listed them all. I suggest the authors proofread their manuscript more carefully and correct all the mistake thoroughly in the text.

RESPONSE TO COMMENT 1:

Thank you for appreciating our manuscript and for carefully revision. In accordance with suggestion, we have carefully reviewed and corrected the inconsistencies throughout the whole text. The references change is visible in the page 7, line 194 [ref. 58 instead of ref. 59];  page 7, line 195 [addition of ref. 58 that was missing]; page 7, line 196 [addition of ref. 59 that was missing]; page 7, lines 205, 208 [addition of ref. 60 that was missing]; page 8, line 215, [ref. 60 instead of ref. 61]; page 8, line 230 [addition of ref. 64 that was missing]; page 8, line 231 [ref. 65 instead of ref. 64];  page 8, line 232 [the cited author ar not Sholhui but Park ref. 65]; page 8, line 234 [addition of ref. 65 that was missing]; page 9, line 258 [addition of ref. 75 that was missing]; page 9, line 271 [addition of ref. 76 that was missing]; page 12, lines 347, 350  [addition of ref. 91 that was missing]; page 12, lines 358, 364 [addition of ref. 89 that was missing], page 12, line 365 [refs. 81,84 insert after “as reporting in other studies”]; page 12, line 372 [addition of ref. 93 that was missing]; page 13, lines 375, 380, 384 [addition of ref. 85 that was missing]; pages 13, 14, lines 401, 408, [addition of ref. 83 that was missing]; page 14, line 421 [addition of ref. 84 that was missing]; page 16, line 468 [addition of ref. 106 that was missing]; page 16, line 474 [addition of ref. 107 that was missing]; page 16, line 490 [addition of ref. 109 that was missing]; page 17, line 507 [addition of ref. 110 to 111 that was missing]; page 19, line 583 [addition of ref. 123 that was missing]; page 20, line 598 [addition of ref. 124 that was missing] and page 20, line 616 [addition of ref. 125 that was missing]. Moreover, we have corrected some grammatical and formatting mistakes that we found in the text.

COMMENT 2:

In figure 2, the authors used multiple circles to represent different cardiometabolic risk factors that related with GA. Does the size of the circle indicate the importance of GA in correlating with different risk factors? If yes, please provide detailed description with supporting reference. If not, please change the circle to the same size.

RESPONSE TO COMMENT 2:

The size of the circle does not indicate the importance of GA in correlation with different risk factors. Unfortunately, from literature data, we are unable to establish a definitive greater or lesser contribution of GA on each risk factors taken into consideration. GA has all the credentials to be an excellent predictor of cardio-metabolic risk if evaluated in its fluctuations compared to the metabolic parameters, although additional studies though are needed to introduce GA in clinical practice and guidelines for diagnosis and monitoring of therapies in patients affected by cardio-metabolic disorders. We are thankful for the careful observation. As requested, we have changed the circle to the same size (Figure 2 in the revised manuscript, page 6).

Reviewer 2 Report

The manuscript (review) submitted by Giglio and colleagues summarize the vast data available from published literature regarding the significance of Glycated Albumin (GA) for diagnosis of Diabetes and its complications including the Kidney, Cerebro- and cardiovascular diseases. The discussed topics in the manuscript are very important for the diabetes research field. Although the authors have put efforts in summarizing the available data, the manuscript lacks coherence and there are some major issues (as detailed below) that needs to be addressed. Hence the manuscript is not recommended for publication in the current form.

Comments:

  1. The authors explain the three-domains structure of the albumin under the sub-heading Glycated Albumin (lines 71-74). The lines 71-72 are not very clear. It would be better if they could rephrase it to make it clear.
  2. The authors should make the Figure 1 little big and increase the font size to make it legible. Also, the notes on the top of the figure can be made as table and increase the font size for that too. Currently the figure 1 is not readable.
  3. Figure 2.: Explain briefly what does the figure mean.
  4. Line 152: Author means Insulin Failure right instead of insular failure?
  5. In the section 4.1 GA and Diabetes, the authors refer to only T2D but not T1D. Can the authors comment on that? Moreover, the authors mention that since the glycation rate is greater for Albumin compared to HbA1C, it would reflect early hyperglycemia. Do the authors have any clear time difference between both the processes, such as GA is observed few weeks in advance of HbA1C. Having a specific time is always better than arbitrary ones.
  6. Lines 168-170 under section 4.1.1 need to be rephrased. It is not clear now.
  7. Lines 176- 178 under section 4.1.2 is not clear. Do the authors mean some patients show significant reduction in blood glucose from the start of the treatment while others show only after 6 months? Currently, the statement does not flow well.
  8. Lines 180-184. Under section 4.1.2, the authors need to split the sentence into either 2 or 3. Currently, it is put together into 1 sentence and it is very confusing for the readers.
  9. Paragraph 2 under section 4.1.2 comparing the HbA1C and GA levels in determining the short-term variations of glycemic control is hard to understand. The authors should try to make it more clear and probably limit using details provided which might be confusing for the readers.
  10. Like the HbA1C which has a range to determine the normal, prediabetic and diabetic, does GA has a range for classifying patients for normal, prediabetic and diabetic. Also, for the other metabolic syndrome related parameters as well. Is there a range of GA that determines the risk for the above mentioned parameters.
  11. Section 4.2.2 has to be reformatted and rewritten with less information. At present it has a lot of information about different studies. Although, it is very important to mention these studies and signify their importance, it is not required to elaborate on the details. Also, the authors mention GA correlation only in the last paragraph of the section which seems separated from the entire section. Instead the authors should reduce the details given about the studies and try to just correlate the GA levels where needed.
  12. Lines 349 and 350 mention about the GA % in prediabetes and Diabetes. However, is there any range for the GA % to classify them in the above 2 groups published already? That would be more useful.
  13. Section 4.3.1 also has lot of information on the different studies that correlated GA levels with BMI or obesity. However, it is important to make a summary of all these studies and not provide a lot of details. All the sections should be concise and explain the important findings from different studies and end with the authors perspective or the interpretation which would help the reader.
  14. I really like the idea of using a table showing all the different studies and correlation of metabolic risk factor with GA cut off. Only one suggestion regarding this would be to have a clear demarcation between the different risk factors. 
  15. Overall, the authors have done a tremendous job in putting together a lot of information on GA and its relevance in metabolic risk factors as a biomarker. However, the authors need to address the above mentioned comments to make it concise and clear for the readers.

Author Response

REVIEWER #2:

The manuscript (review) submitted by Giglio and colleagues summarize the vast data available from published literature regarding the significance of Glycated Albumin (GA) for diagnosis of Diabetes and its complications including the Kidney, Cerebro- and cardiovascular diseases. The discussed topics in the manuscript are very important for the diabetes research field. Although the authors have put efforts in summarizing the available data, the manuscript lacks coherence and there are some major issues (as detailed below) that needs to be addressed. Hence the manuscript is not recommended for publication in the current form.

We would like to express our great appreciation to reviewer for the valuable comments and suggestions that allowed us to improve our manuscript.

COMMENT 1:

The authors explain the three-domains structure of the albumin under the sub-heading Glycated Albumin (lines 71-74). The lines 71-72 are not very clear. It would be better if they could rephrase it to make it clear.

RESPONSE TO COMMENT 1:

We have rephrased it as following, pages 2-3, lines 71-75: “Albumin is the most abundant plasma protein, with a half-life of about 21 days, that undergoes non-enzymatic glycosylation due to its high circulating concentration, and the number of lysine and arginine residues making it more prone to glycation [6,25]. The protein consists of 585 amino acids residues organized in a single polypeptide chain stabilized by 17 disulphide bridges and comprising 3 homologous domains (I, II, and III) assembled to form a heart-shaped molecule. Each domain is further organized into 2 subdomains (A and B), which share analogous structural motifs. The glycation-induced modifications significantly alter the structure of the two subdomains of the second domain. These two subdomains act as binding sites, with preferential activity for different endogenous substrates  (such as bilirubin and porphyrins), and exogenous compounds (like benzodiazepines and ibuprofen)”.

COMMENT 2:

The authors should make the Figure 1 little big and increase the font size to make it legible. Also, the notes on the top of the figure can be made as table and increase the font size for that too. Currently the figure 1 is not readable.

RESPONSE TO COMMENT 2:

Thank you for this comment. We made the changes as kindly recommended, making Figure more readable (Figure 1 in the revised manuscript insert in the text, page 3).

COMMENT 3:

Figure 2.: Explain briefly what does the figure mean.

RESPONSE TO COMMENT 3:

Figure 2 schematically represents the correlation between GA and different cardiometabolic risk factors such as diabetes, dyslipidaemia, obesity, kidney disease and cerebro-cardiocascular diseases. Furthermore, these risk factors have been divided in two groups based on their negative or positive correlation with GA.

Consequently, we have implemented the title of the Figure 2 as following (page 6, lines 159-161): “Schematic representation of correlations between GA and different metabolic and vascular parameters. The citated parameters are divided into two group based on their negative or positive correlation with GA and report the same colour as the risk factor to which they refer”.

COMMENT 4:

Line 152: Author means Insulin Failure right instead of insular failure?

RESPONSE TO COMMENT 4:

We have corrected replaced the term and reformulated the phrase as following (page 5, lines 150-152): “The pathogenesis of Type 2 Diabetes Mellitus (T2DM) is linked to the insulin-resistance, which leads to slow decline of pancreatic β-cells; both pathological states influence each other and presumably synergistically exacerbate diabetes inducing the failure insulin over time [51]”.

COMMENT 5:

  1. In the section 4.1 GA and Diabetes, the authors refer to only T2D but not T1D. Can the authors comment on that?
  2. Moreover, the authors mention that since the glycation rate is greater for Albumin compared to HbA1C, it would reflect early hyperglycemia. Do the authors have any clear time difference between both the processes, such as GA is observed few weeks in advance of HbA1C. Having a specific time is always better than arbitrary ones.

RESPONSE TO COMMENT 5:

  1. The section 4.1 with the title “GA and Diabetes” is generic. Although in the literature there are several articles on GA and the different forms of diabetes, we have chosen to be particularly focused on type 2 diabetes mellitus because it is commonly associated with overweight, obesity and other metabolic disorders. The aim of this review is to describe the GA’s contribution in the diagnosis and monitoring of Type 2 diabetes mellitus and T2DM-related metabolic conditions as well as to present its potential usefulness for predicting and stratifying the overall cardiometabolic risk.
  2. Because of high albumin accessibility, high concentration and half-life, its glycation is greater than that of hemoglobin. It is known that glycation rate of albumin depends on glycemia and the time the albumin stays in the bloodstream, so GA has been proposed to be a biomarker of glycemic status. Additionally, due to the shorter lifespan of albumin in comparison to the traditional biomarkers of glycemic control, like HbA1c, GA can be considered as a biomarker of early response to hypoglycemic treatment. However, there is not accurate time difference between both the processes; the GA test provides information on the mean blood glucose concentration of the last 15-20 days before blood collection, while HbA1C of the last 3 months as reporting in the original text.

COMMENT 6:

Lines 168-170 under section 4.1.1 need to be rephrased. It is not clear now.

RESPONSE TO COMMENT 6:

To clarify this concept, we have reworded the sentence as following (page 6, lines 173-177).

“The homeostasis model assessment of β-cell function (HOMA-β), an index of insulin secretory function, is negatively correlated with GA but it is not correlated with HbA1c. In fact, it has been assumed that this phenomenon occurs due to the association between reduced insulin secretion and increased glucose excursions including postprandial hyperglycemia. Therefore, GA levels reflect glucose excursions and HOMA-β plays a central role in the GA regulation mechanism [49]”.

COMMENT 7:

Lines 176-178 under section 4.1.2 is not clear. Do the authors mean some patients show significant reduction in blood glucose from the start of the treatment while others show only after 6 months? Currently, the statement does not flow well.

RESPONSE TO COMMENT 7:

As kindly required, we have rewritten the sentence (page 7, lines 183-186).

“The time of the therapy efficacy is significantly different: some patients do not show a significant reduction in blood glucose from the beginning of treatment, but at least after six months [17], requiring a remodulation/change of therapy in a short time in order to avoid the consequences of a glycemic failure [16,18,19]”.

COMMENT 8:

Lines 180-184. Under section 4.1.2, the authors need to split the sentence into either 2 or 3. Currently, it is put together into 1 sentence and it is very confusing for the readers.

RESPONSE TO COMMENT 8:

As recommended, we have simplified the phrase as following (page 7, lines 187-192).

“GA levels may be useful also in predicting the patient's response to hypoglycemic therapy. In particular, Lee et al. performed a study including Korean patients > 40 years of age with poorly controlled insulin-naïve T2DM. These subjects, treated with glimepiride (sulphonylurea (SU)), were randomized based on obesity and the dose of glimepiride with a 1:1 ratio of insulin detemir (prolonged-action analog) and Biphasic Insulin (BIA) (70% insulin aspart protamine and 30% insulin aspart Analogue) [58]”.

COMMENT 9:

Paragraph 2 under section 4.1.2 comparing the HbA1C and GA levels in determining the short-term variations of glycemic control is hard to understand. The authors should try to make it more clear and probably limit using details provided which might be confusing for the readers.

RESPONSE TO COMMENT 9:

In order to make it more clear, we have modified the section 4.1.2. as following (page 9, lines 193-216).

“Patients who failed to achieve <20% GA at 3 weeks were switched to BIA twice a day for 16 weeks [58]. Mean HbA1c, GA, fasting and stimulated plasma glucose levels were significantly reduced after 16 weeks (p< 0.001), and 40% of patients reached the target HbA1c (<7%) [58]. GA decreased more rapidly than HbA1c during intensive insulin therapy (p= 0.0004) [59]. Thus, GA has been considered as a useful marker for monitoring short-term variations of glycemic control during the treatment of diabetic patients [58,59].

GA could be also potentially useful for monitoring the beneficial effects of nutraceuticals. Li et al. performed a double-blind, randomized, placebo-controlled study to evaluate fiber supplementation during a 12-week integration period and measured a complete set of glucose metabolism and lipid biomarkers [60]. The aim of the study was to determine the effects of a soluble fiber dietary supplement with dextrin (NUTRIOSE®, Roquette Frères, Lestrem, France) on insulin resistance and on the determinants of metabolic syndrome (MeS) in overweight Chinese men (age= 30.4±4.3 years; BMI= 24.5±0.2 kg/m2). All glucose metabolism biomarkers (HOMA-Insulin Resistance, HbA1C and GA) improved after 12 weeks in the test group, showing an increase of adiponectin concentrations (p= 0.05) and a significant reduction of glucose, insulin, HOMA-Insulin Resistance, HbA1C and GA (all p< 0.01) [60].

Furthermore, lipid metabolism biomarkers (TG, TC, LDL-C, VLDL-C and HDL-C) also improved. The intake of dietary fiber helps to improve the glucose metabolism by releasing some intestinal peptides (such as glucagon-like peptide-1) [61]. The proposed mechanism for the described effects of NUTRIOSE® was linked also to the release of adiponectin, a well-known cytokine secreted by adipose tissue, which regulates glucose metabolism, and is able to stimulate the oxidation of fatty acids, lowering plasma triglycerides and improving insulin sensitivity. As improvements in glucose levels was associated with a reduction in GA levels [60], GA has been suggested to be a useful tool for evaluating the response to the dietary fiber treatment”.

COMMENT 10:

Like the HbA1C which has a range to determine the normal, prediabetic and diabetic, does GA has a range for classifying patients for normal, prediabetic and diabetic. Also, for the other metabolic syndrome related parameters as well. Is there a range of GA that determines the risk for the above mentioned parameters.

RESPONSE TO COMMENT 10:

It seems that a GA value greater than 17.9% is predictive for cardiovascular events in cardiometabolic patients. We preferred to not to give it as a confirmed cut-off because there is no consensus on the decisional values to use for the different mentioned parameters (see Table 1). Our review summarizes the currently available knowledge and should making suggestions for future studies.

COMMENT 11:

Section 4.2.2 has to be reformatted and rewritten with less information. At present it has a lot of information about different studies. Although, it is very important to mention these studies and signify their importance, it is not required to elaborate on the details. Also, the authors mention GA correlation only in the last paragraph of the section which seems separated from the entire section. Instead the authors should reduce the details given about the studies and try to just correlate the GA levels where needed.

RESPONSE TO COMMENT 11:

We have tried to rewrite this section including less details in order to make it more readable: (pages 8-10, lines 237-288).

“Patients with dyslipidemia who do not reach the LDL-C target with lifestyle changes only and nutraceuticals, were undergoing to statin treatment. Statins are drugs that act on the 3-hydroxy-3-methyl-glutaryl-coenzyme A reductase (HMG-CoA Reductase), the enzyme responsible for the modulation of cholesterol synthesis in the liver. The statins have numerous pleiotropic effects, including CVD benefit, but also recently it has been widely reported for incretins, innovative anti-diabetes drugs.

Despite the beneficial effects of atorvastatin and rosuvastatin on the levels of lipoprotein, cholesterol [66-69] and C-reactive protein (CRP), there are some important concerns about the effects of statins on glucose homeostasis. Statin intolerance (SI) syndrome is not only characterized by the increase of creatine-phospho kinase (CPK) and other bio humoral biomarkers (such as lipase and amylase), but also can induce the development of diabetes [70,71]. That has been confirmed by a meta-analysis by Sattar et al. [72]. Moreover, studies on the impact of statins (rosuvastatin and pravastatin) on glucose, as the Justification for the Use of statins in Prevention: an Intervention Trial Evaluating Rosuvastatin (JUPITER) study and PRavastatin Or atorVastatin Evaluation and Infection Therapy–Thrombolysis In Myocardial Infarction (PROVE-IT TIMI 22) study, gave more information on this field of research [73,74].

A subgroup of the study performed by Thongtang et al. [75] included non-pregnant women and men with hypercholesterolaemia [adults aged 18 years and over; LDL-C≥ 160 mg/dl (4.1 mmol/l) and <250 mg/dl (6.5 mmol/l) and TG<400 mg/dl (4.5 mmol/l)]; they were randomized to different doses of statin and were asked to follow a National Cholesterol Education Program (NCEP) step 1 diet for 6 weeks. After 4 and 6 weeks, a blood sample was collected to measure lipid and lipoprotein parameters (TC, TG, calculated LDL-C, HDL-C, Apo AI and Apo B measurements) [75]. While atorvastatin at the maximum dose increased GA levels by 0.8% from baseline, rosuvastatin slightly decreased GA levels by 0.7% only after 6 weeks of treatment in hyperlipidemic subjects compared to the JUPITER study (p= 0.002). Rosuvastatin 20 mg/day caused a small but significant increase in HbA1c levels in normolipidemic subjects with high CRP [75].

A randomized, double-blind study examined the safety of ezetimibe compared to placebo for change from baseline to 24 weeks in HbA1c (primary endpoint), GA and FPG (secondary endpoints) in Japanese subjects with T2DM and hypercholesterolemia [76]. One hundred fifty-two adult patients with T2DM and hypercholesterolemia, whose LDL-C measured <160 mg/dl (subjects taking lipid-lowering drugs) or <140 mg/dl (subjects who do not take lipid-lowering drugs) at the beginning of the screening phase, were randomized after a 5-week wash-out period to ezetimibe 10 mg or placebo (1:1) for 24 weeks. Changes in HbA1c, GA and fasting glycemia from baseline to week 24 were evaluated. After this treatment period, HbA1c increased significantly in both ezetimibe and placebo groups (the difference between treatments 0.08 [95% CI: -0.07 to 0.23]) [76]. At 24 weeks, the mean change from baseline in GA levels (difference between treatments 0.00 [95% CI: -0.47, 0.47]) and FPG (the difference between treatments −4.8 [95% CI: -12.1, 2.1]) were similar in both groups. So, GA does not change significantly after ezetimibe therapy [76], contrary to what occurs after statin treatment [65,75]. A randomized, double-blind, placebo-controlled study was conducted in adult patients with T2DM (HbA1C> 7.5% [58 mmol/mol]) treated with metformin and hypertriglyceridemia (TG> 200 and <500 mg/dL) to determine the effects of volanesorsen (acronym ISIS 304801, a second-generation chimeric antisense 2'-O-methoxyethyl chimeric Apo C-III inhibitor) on TG levels and IR in patients with T2DM [77]. Patients were randomized 2:1 to receive volanesorsen 300 mg or placebo for a total of 15 subcutaneous weekly doses. Volanesorsen treatment significantly reduced the plasma levels of ApoC-III (288%, p= 0.02) and TG (269%, p= 0.02) and increased HDL-C (42%, p= 0.03) compared to placebo. These changes were accompanied by a 57% improvement in whole-body insulin sensitivity (p< 0.001). Better insulin sensitivity was sufficient to significantly reduce GA (14.1%, p= 0.034) and fructosamine (238.7 mmol/L, p= 0.045) at the end of the treatment and HbA1c (44%, p= 0.025) 3 months after the treatment [77]. These data suggest that GA improved after cholesterol-lowering treatments without change the dose of hypoglycemic therapy and that GA could potentially be used for monitoring the efficacy of lipid-lowering therapies, except ezetimibe [65]”.

COMMENT 12:

Lines 349 and 350 mention about the GA % in prediabetes and Diabetes. However, is there any range for the GA % to classify them in the above 2 groups published already? That would be more useful.

RESPONSE TO COMMENT 12:

As reporting in the text, the GA Upper Range Limit (URL) of 14.5% (95% CI: 14.3-14.7) has been established in Caucasian healthy subjects. GA cut-off of 15.5% (corresponding to an HbA1c level of 5.8%) could be optimal for predicting diabetes. In diabetic patients the levels of GA start to 16.7% (corresponding to an HbA1c level of 6.6%). These values can vary with the presence of other risk factors associated with diabetes. However, these are guideline values obtained from the studies currently present in the literature, but it must be confirmed by future studies in order to introduce the measurement of GA in clinical practice

COMMENT 13:

Section 4.3.1 also has lot of information on the different studies that correlated GA levels with BMI or obesity. However, it is important to make a summary of all these studies and not provide a lot of details. All the sections should be concise and explain the important findings from different studies and end with the authors perspective or the interpretation which would help the reader.

RESPONSE TO COMMENT 13:

As suggest, we have modified the text and provided a new Table that summarizes the data on indications and limits of Glycated Albumin in clinical practice with reference to various cardiometabolic risk factors to help the reader obtain a better understanding (page 10-13, lines 305-387; see also Table 2).

“Weight variation is associated with a high rate of prediabetes in obese patients. Both glycated proteins (i.e., HbA1c and GA) are known to be influenced by various factors other than the glycemia; the GA value, for example, is negatively associated with the BMI in adult diabetic patients. However, the connection between obesity and GA remains to be fully explored.

Number of studies have been focused on the negative relationship between BMI and GA in the diabetic population [83-85]. He et al. [85] found a negative relationship between BMI and GA in diabetes, as each 1 kg/m2 increase in BMI corresponded to a 0.1% decrease in the absolute value of GA. The negative relationship between GA and BMI is probably a consequence of the inverse correlation between BMI and total albumin. The latter is probably due to the weight loss, which leads to kidney hyperfiltration decreases, associated with a decrease in estimated glomerular filtration rate (eGFR), causing the decrease of total albumin urinary clearance and the increase of circulating albumin [86]. Obese subjects usually have a lower serum albumin concentration than non-obese, and the fact that GA is a ketamine formed through the non-enzymatic glycation reaction of serum albumin partly explains the relatively low concentration of GA in obese patients with diabetes [87].

In T2DM patients, BMI is associated with an abnormal albumin metabolism compared to patients without T2DM. Moreover, the absence of a correlation between CRP and GA in patients with T2DM suggests that an increase in albumin metabolism due to chronic inflammation is unlikely. BMI regulates insulin secretion in T2DM patients and insulin secretion regulates GA through its effects on glucose excursions and/or postprandial hyperglycemia [84].

However, the inflammation proteins that are related to obesity [88] are probably associated with changes in GA levels. Even inflammation plays a determining role in the inverse link between BMI and GA; in fact, chronic inflammation in obesity can increase the albumin catabolism rate and decrease the synthesis rate, leading to an increase in the turnover of serum albumin in obese subjects; for the same glucose concentration, obese subjects may have a lower serum GA concentration than non-obese subjects [89].

A reduced GA value is associated with the increase in waist circumference (WC) values in patients with hyperglycemia, demonstrating significant implications for the clinical application of GA in monitoring blood glucose. One thousand seven hundred ninety-nine subjects with central obesity had lower GA levels and GA/HbA1c ratio than those without (both p< 0.01). GA levels and GA/HbA1c ratio were negatively correlated with central obesity (both p< 0.01), while HbA1c was not correlated (p= 0.833). In euglycemic and hyperglycemic subgroups, GA and GA/HbA1c ratio showed decreasing trends with increasing WC levels (both p< 0.01). WC was a significant negative determinant of GA (p< 0.05). In hyperglycemic subjects, the GA value decreased by 0.15% for every 5 cm increase in the WC, regardless of the presence of central obesity [90].

In a study of Zemlin et al., on 1294 African adults, obesity was more pronounced in the diabetes and prediabetes groups detected with the mean BMI screen of 32.5 kg/m2 and 31.5 kg/m2, respectively. The optimal GA thresholds for diagnosing diabetes and prediabetes detected by the screen were 14.90% and 12.75%, respectively [57].

Sumner et al. assessed the ability of HbA1c and GA to detect prediabetes in non-obese (BMI within 30 kg/m2) and obese (BMI greater than 30 kg/m2) by OGTT on 236 healthy African immigrants (BMI 27.6±4.4 kg/m2) [91]. BMI and HbA1c were positively correlated (r= 0.22, p< 0.001), while BMI and GA were negatively correlated (r= -0.24, p< 0.001) [91]. Although the sensitivity of HbA1c and GA were similar in non-obese immigrants (37% vs 42%, p= 0.75), prediabetes was detected in about 9% non-obese Africans by GA alone, in about 7% by HbA1c alone and only in 2% by both tests [91]. In obese subjects, the HbA1c was a much better diagnostic test than the GA and the combination of the tests did not improve the sensitivity [91,92].

Koga et al. enrolled non-diabetic subjects [51.8 y (range, 28-78 y); 158 with normal glucose tolerance and 54 with reduced glucose tolerance; BMI 23.6 kg/m2 (range, 16.2-37.5 kg/m2)] for 3 weeks and analyzed the effects of FPG, 2 hour OGTT, age, BMI and hsCRP on HbA1c and on GA [89]. In these subjects, a correlation between HbA1c and GA was found (r= 0.361, p< 0.001); FPG was correlated with HbA1c (r= 0.455, p< 0.0001) and weakly correlated with GA (r= 0.181, p< 0.01). BMI showed a positive correlation with HbA1c (r= 0.280, p< 0.0001) and a negative correlation with GA (r= −0.303, p< 0.0001) [89]. Furthermore, multivariate analysis showed that FPG (r= 0.353, p <0.0001), OGTT 2-h glucose (r= 0.213, p= 0.001) and age (r= 0.178, p= 0.004) were independently associated with HbA1c, whereas FPG (r= 0.278, p< 0.001) and age (r= 0.224, p= 0.001) were positively linked. BMI was negatively associated with GA (r= −0.419, p <0.0001). The GA/HbA1c ratio was negatively associated with BMI (r= −0.499, p< 0.0001). In this study, the inflammatory status associated with obesity was also assessed by measuring plasma hsCRP, which was associated positively with BMI (r= 0.458, p <0.0001) and negatively with GA (r= −0.300, p< 0.0001) [89] as reporting in other studies [81,84].

Reynolds et al. examined young euglycemics (21.1±3.9 years, BMI 23.9±4.0 kg/m2) mainly of European origin for 1 month by subjecting them to a fasting test of 50g of carbohydrates in order to explore the relationship among GA, fasting and 2-hour post-load blood glucose measurements, by the incremental area under the glucose curve, glycemic interval, BMI and CRP [93]. Mean blood glucose values (FPG 4.7± 0.5 mmol/L, 2h PG 5.3±0.8 mmol/L, GA 11.7±1.3 %) suggest a normal glucose tolerance. Moreover, circulating albumin was not related to the postprandial glycemic response in young euglycemic adults [93].

He et al. enrolled 1177 men and 1385 women (51±13 years) to evaluate the relationship between BMI and GA levels [85]. In diabetic patients (n=1223), the levels of GA, HbA1c and GA/HbA1c ratio were 16.7±3.0%, 6.6±0.9% (49±9 mmol/mol) and 2.51±0.33, respectively [85]. In non-diabetic subjects (n= 1339), the concentrations of GA, HbA1c and GA/HbA1c ratio were 13.8±1.7%, 5.6±0.4% (38±4 mmol/mol) and 2.47±0.31, respectively. Decreasing trends in GA levels and GA/HbA1c ratio and an increasing trend in HbA1c concentration (all p< 0.05) were found linked to the increase in BMI, regardless of diabetes status. Multiple regression analysis revealed that the BMI was independently related to HbA1c in the non-diabetic population (r= 0.158, p< 0.001) [85]. BMI had a negative effect on the GA/HbA1c ratio, as BMI was negatively related to GA concentrations in both diabetic and non-diabetic patients. However, it was not related to HbA1c in the diabetic population and was positively associated with HbA1c in the non-diabetic population, which can explain the inverse relationship between BMI and the GA/HbA1c ratio [85]. This literature evidence shows that the increase of BMI reflects a decrease of GA levels even in the presence of diabetes. In summary, in obese diabetic patients, chronic inflammation associated with obesity could increase albumin metabolism and negatively regulate GA levels [81,84,85]”.

COMMENT 14:

I really like the idea of using a table showing all the different studies and correlation of metabolic risk factor with GA cut off. Only one suggestion regarding this would be to have a clear demarcation between the different risk factors.

RESPONSE TO COMMENT 14:

As kindly suggested, we have graphically modified the Table 1 in order to have a clear demarcation between the different risk factors (see Table 1).

COMMENT 15:

Overall, the authors have done a tremendous job in putting together a lot of information on GA and its relevance in metabolic risk factors as a biomarker. However, the authors need to address the above mentioned comments to make it concise and clear for the readers.

RESPONSE TO COMMENT 15:

Hopefully, we have responded adequately to all your valuable comments. Thanks for appreciating the work done. The data in this topic is still scarce and does not allow to draw a definitive conclusion. We hope that the revised manuscript is made clearer and more concise compared to the original submission.

Reviewer 3 Report

This review articles is discussing about glycated albumin as the potential biomarker for metabolic risk factor in which HbA1c not suitable for marker. Authors have reviewed extensively 74 articles and provided their conclusions in this review article.  It is heavily filled with data and conclusion from all the sources they are using. 

My comments and suggestions for the authors:

  1. I feel this article is difficult to read and to digest.  It is hard for me to get the message.  I guess it is because there are multiple ideas in 1 sentence throughout the article.  I think the authors need to simplify every sentence and avoid redundant words.  The other things that the authors need to organize their thought better. 
  2. The authors tend to put all the information that they read from the paper into this review article, make this review article is heavy with primary data in the text.  I think the readers can get all primary data from the original paper, however, the readers need to get the authors' impression about the primary data, which is lacking in this review article. 
  3. I like the fact that authors making the summary Table of the articles they reviewed. However, this Table is difficult to follow, there is not clear separation between 1 study to the other study.  The authors need to work more on the Table to make it clearly separated and give the main conclusion only. 

Author Response

REVIEWER #3:

This review articles is discussing about glycated albumin as the potential biomarker for metabolic risk factor in which HbA1c not suitable for marker. Authors have reviewed extensively 74 articles and provided their conclusions in this review article. It is heavily filled with data and conclusion from all the sources they are using.

Thank for your revision. We have responded to your comments, trying to give specific meaning to the conclusions from different sources mentioned in our work through a new table. All changes in the text are highlighted in red color.

COMMENT 1:

I feel this article is difficult to read and to digest.  It is hard for me to get the message. I guess it is because there are multiple ideas in 1 sentence throughout the article. I think the authors need to simplify every sentence and avoid redundant words. The other things that the authors need to organize their thought better.

RESPONSE TO COMMENT 1:

We have edited our text to clarify the message, in particularly we have simplified the sentences and rephrased the text where need (pages 2,5-13).

COMMENT 2:

The authors tend to put all the information that they read from the paper into this review article, make this review article is heavy with primary data in the text. I think the readers can get all primary data from the original paper, however, the readers need to get the authors' impression about the primary data, which is lacking in this review article.

RESPONSE TO COMMENT 2:

It is true that we tended to put all the information currently available, in order to make a single article with all the details on this topic, allowing to the reader to get an idea about GA and every single risk factor, considering variables such as age, race, gender, therapies that have been used, their impact as well as, treatments’ length, design of the clinical studies performed including concomitant diseases and more since there is still no consensus on the reference cut-off and possible use of GA in clinical practice as a tool for cardiometabolic risk assessment. This article should serve as a starting point for the readers to try in a future studies to confirm the efficacy of this biomarker with a promising potential in the predicting of cardiovascular disease regardless of its role in diabetes. However, we agree that some parts were difficult to follow and as kindly required also by other reviewers, we have modified the text making it clearer and more concise. Also, we have made a new table (see Table 2 in the revised manuscript) summarizing the main conclusions on every risk factor related to GA.

COMMENT 3:

I like the fact that authors making the summary Table of the articles they reviewed. However, this Table is difficult to follow, there is not clear separation between 1 study to the other study.  The authors need to work more on the Table to make it clearly separated and give the main conclusion only.

RESPONSE TO COMMENT 3:

As you kindly requested, Table 1 has been modified including a clear demarcation between the different risk factors. Furthermore, as specified in the previous comment, we have made also a new Table 2 where on indications and limits of Glycated Albumin in clinical practice with reference to various cardiometabolic risk factors.

Round 2

Reviewer 1 Report

The authors have addressed all my comments. I suggest to accept the manuscript in present form.

Author Response

We would like to thanks  the Expert Reviewer for the acceptance of our revised ms, which is surely improved through the suggestions and comments. Thanks again

Prof. F Mannello

Reviewer 2 Report

The authors have elaborated most of the sections in the review with too many details that are not required and helpful to the readers.

COMMENT 4:

Line 152: Author means Insulin Failure right instead of insular failure?

RESPONSE TO COMMENT 4:

We have corrected replaced the term and reformulated the phrase as following (page 5, lines 150-152): “The pathogenesis of Type 2 Diabetes Mellitus (T2DM) is linked to the insulin-resistance, which leads to slow decline of pancreatic β-cells; both pathological states influence each other and presumably synergistically exacerbate diabetes inducing the failure insulin over time [51]”.

inducing the failure of insulin over time [51]”.

COMMENT 7:

Lines 176-178 under section 4.1.2 is not clear. Do the authors mean some patients show significant reduction in blood glucose from the start of the treatment while others show only after 6 months? Currently, the statement does not flow well.

RESPONSE TO COMMENT 7:

As kindly required, we have rewritten the sentence (page 7, lines 183-186).

“The time of the therapy efficacy is significantly different: some patients do not show a significant reduction in blood glucose from the beginning of treatment, but at least after six months [17], requiring a remodulation/change of therapy in a short time in order to avoid the consequences of a glycemic failure [16,18,19]”.

Still not clear.

COMMENT 11:

Section 4.2.2 has to be reformatted and rewritten with less information. At present it has a lot of information about different studies. Although, it is very important to mention these studies and signify their importance, it is not required to elaborate on the details. Also, the authors mention GA correlation only in the last paragraph of the section which seems separated from the entire section. Instead the authors should reduce the details given about the studies and try to just correlate the GA levels where needed.

RESPONSE TO COMMENT 11:

We have tried to rewrite this section including less details in order to make it more readable: (pages 8-10, lines 237-288).

This section is still very long and include too many details which are not required.

COMMENT 13:

Section 4.3.1 also has lot of information on the different studies that correlated GA levels with BMI or obesity. However, it is important to make a summary of all these studies and not provide a lot of details. All the sections should be concise and explain the important findings from different studies and end with the authors perspective or the interpretation which would help the reader.

RESPONSE TO COMMENT 13:

As suggest, we have modified the text and provided a new Table that summarizes the data on indications and limits of Glycated Albumin in clinical practice with reference to various cardiometabolic risk factors to help the reader obtain a better understanding (page 10-13, lines 305-387; see also Table 2).

Most of the sections in the review are too long because of the details provided by the authors. The authors have not addressed these points.

Author Response

First of all, we would like to thanks the Expert Reviewer for the constructive attention to our ms and for the insightful suggestions.

 Please, find, point-by-point replies to your comments:

The authors have elaborated most of the sections in the review with too many details that are not required and helpful to the readers.

R: We agree with the Reviewer; our details are focused to improve clarity and enhance readibility with insightful informations.

COMMENT 4:

Line 152: Author means Insulin Failure right instead of insular failure?

RESPONSE TO COMMENT 4:

We have corrected replaced the term and reformulated the phrase as following (page 5, lines 150-152): “The pathogenesis of Type 2 Diabetes Mellitus (T2DM) is linked to the insulin-resistance, which leads to slow decline of pancreatic β-cells; both pathological states influence each other and presumably synergistically exacerbate diabetes inducing the failure insulin over time [51]”.

inducing the failure of insulin over time [51]”.

R: We agree with the correction of Reviewer; sorry for the mistake.  we modified the phrase accordingly: "inducing the failure of insulin over time [51]"

COMMENT 7:

Lines 176-178 under section 4.1.2 is not clear. Do the authors mean some patients show significant reduction in blood glucose from the start of the treatment while others show only after 6 months? Currently, the statement does not flow well.

RESPONSE TO COMMENT 7:

As kindly required, we have rewritten the sentence (page 7, lines 183-186).

“The time of the therapy efficacy is significantly different: some patients do not show a significant reduction in blood glucose from the beginning of treatment, but at least after six months [17], requiring a remodulation/change of therapy in a short time in order to avoid the consequences of a glycemic failure [16,18,19]”.

Still not clear.

R: We  rewrite accordingly the phrase: "The time of the therapy efficacy is significantly different among patients: in particular, some patients do not show a significant reduction in blood glucose from the  start of the treatment treatment, but only  after six months later [17], suggesting a requirement of a remodulation/change of therapy in a more short time to avoid the consequences of a glycemic failure [16,18,19]"

COMMENT 11:

Section 4.2.2 has to be reformatted and rewritten with less information. At present it has a lot of information about different studies. Although, it is very important to mention these studies and signify their importance, it is not required to elaborate on the details. Also, the authors mention GA correlation only in the last paragraph of the section which seems separated from the entire section. Instead the authors should reduce the details given about the studies and try to just correlate the GA levels where needed.

RESPONSE TO COMMENT 11:

We have tried to rewrite this section including less details in order to make it more readable: (pages 8-10, lines 237-288).

This section is still very long and include too many details which are not required.

R: We have shortened the paragraph, focusing attention to not lose important informations,  useful for the Readers.

COMMENT 13:

Section 4.3.1 also has lot of information on the different studies that correlated GA levels with BMI or obesity. However, it is important to make a summary of all these studies and not provide a lot of details. All the sections should be concise and explain the important findings from different studies and end with the authors perspective or the interpretation which would help the reader.

RESPONSE TO COMMENT 13:

As suggest, we have modified the text and provided a new Table that summarizes the data on indications and limits of Glycated Albumin in clinical practice with reference to various cardiometabolic risk factors to help the reader obtain a better understanding (page 10-13, lines 305-387; see also Table 2).

Most of the sections in the review are too long because of the details provided by the authors. The authors have not addressed these points.

R: We have shortened as possible the paragraphs, focusing attention to more concise infos for improving readibility and understanding of Readers, providing useful perspectives.